Genetic associations of vitamin D receptor polymorphisms with advanced liver fibrosis and response to pegylated interferon-based therapy in chronic hepatitis C

Thanapirom Kessarin 1 2
Suksawatamnuay Sirinporn 1 2
Sukeepaisarnjaroen Wattana 3
Tangkijvanich Pisit 4
Thaimai Panarat 1 2
Wasitthankasem Rujipat 5
Poovorawan Yong 5
http://orcid.org/0000-0002-1357-9547 Komolmit Piyawat 1 2 pkomolmit@yahoo.co.uk
1 Division of Gastroenterology and Hepatology, Department of Medicine, Faculty of Medicine, Chulalongkorn University and King Chulalongkorn Memorial Hospital, Thai Red Cross Society , Bangkok , Thailand
2 Chulalongkorn University, Liver Fibrosis and Cirrhosis Research Unit , Bangkok , Thailand
3 Department of Medicine, Srinagarind Hospital, Faculty of Medicine, Khon Kaen University, Gastroenterology unit , Khon Kaen , Thailand
4 Faculty of Medicine, Chulalongkorn University, Department of Biochemistry , Bangkok , Thailand
5 Department of Pediatrics, Faculty of Medicine, Chulalongkorn University, Center of Excellence in Clinical Virology , Bangkok , Thailand
Zhang Lanjing
Electronic publication date: 2019 Sep 11
Publication date: 2019
Volume: 7
Electronic Location ID: e7666
Received 2019 Apr 30; Accepted 2019 Aug 13
Copyright: © 2019 Thanapirom et al.
Copyright year: 2019
Copyright holder: Thanapirom et al.
License: This is an open access article distributed under the terms of the Creative Commons Attribution License, which permits unrestricted use, distribution, reproduction and adaptation in any medium and for any purpose provided that it is properly attributed. For attribution, the original author(s), title, publication source (PeerJ) and either DOI or URL of the article must be cited.
License URL: https://creativecommons.org/licenses/by/4.0/

Keywords: Vitamin D receptor polymorphisms, Hepatitis C virus, Pegylated interferon, Advanced liver fibrosis

Funding: Ratchadaphiseksomphot Endowment Fund of hepatic fibrosis and cirrhosis research unit GRU 6105530009-1 Ratchadapiseksompotch Fund, Faculty of Medicine, Chulalongkorn University RA59/074 and RA60/101 Thai Association for the Study of the Liver (THASL) Research Chair Grant from the National Science and Technology Development Agency P-15-50004 Center of Excellence in Clinical Virology GCE 59-009-30-005 The study was supported by the Ratchadaphiseksomphot Endowment Fund of hepatic fibrosis and cirrhosis research unit (GRU 6105530009-1), the Ratchadapiseksompotch Fund, Faculty of Medicine, Chulalongkorn University, grant number RA59/074 and RA60/101, the Thai Association for the Study of the Liver (THASL), the Research Chair Grant from the National Science and Technology Development Agency (P-15-50004) and the Center of Excellence in Clinical Virology (GCE 59-009-30-005). There was no additional external funding received for this study. The funders had no role in study design, data collection and analysis, decision to publish, or preparation of the manuscript.

==============================
Vitamin D receptor (VDR) modulates host immune responses to infections such as hepatitis C virus (HCV) infection, including interferon signaling. This study aimed to investigate the associations of VDR polymorphisms with advanced liver fibrosis and response to pegylated interferon (PEG-IFN)-based therapy in patients with chronic HCV infection. In total, 554 Thai patients with chronic HCV infection treated with a PEG-IFN-based regimen were enrolled. Six single-nucleotide polymorphisms (SNPs) were genotyped: the IL28B C > T (rs12979860) SNP and five VDR SNPs, comprising FokI T > C (rs2228570), BsmI C > T (rs1544410), Tru9I G > A (rs757343), ApaI C > A (rs7975232), and TaqI A > G (rs731236). In total, 334 patients (60.3%) achieved sustained virological response (SVR), and 255 patients (46%) were infected with HCV genotype 1. The bAt (CCA) haplotype, consisting of the BsmI rs1544410 C, ApaI rs7975232 C, and TaqI rs731236 A alleles, was associated with poor response (in terms of lack of an SVR) to PEG-IFN-based therapy. The IL28B rs12979860 CT/TT genotypes (OR = 3.44, 95% CI [2.12–5.58], p < 0.001), bAt haplotype (OR = 2.02, 95% CI [1.04–3.91], p = 0.03), pre-treatment serum HCV RNA (logIU/mL; OR = 1.73, 95% CI [1.31–2.28], p < 0.001), advanced liver fibrosis (OR = 1.68, 95% CI [1.10–2.58], p = 0.02), and HCV genotype 1 (OR = 1.59, 95% CI [1.07–2.37], p = 0.02) independently predicted poor response. Patients with the bAt haplotype were more likely to have poor response compared to patients with other haplotypes (41.4% vs 21.9%, p = 0.03). The FokI rs2228570 TT/TC genotypes (OR = 1.63, 95% CI [1.06–2.51], p = 0.03) and age ≥55 years (OR = 2.25; 95% CI [1.54–3.32], p < 0.001) were independently associated with advanced liver fibrosis, assessed based on FIB-4 score >3.25. VDR polymorphisms were not associated with pre-treatment serum HCV RNA. In Thai patients with chronic HCV infection, the bAt haplotype is associated with poor response to PEG-IFN-based therapy, and the FokI rs2228570 TT/TC genotypes are risk factors for advanced liver fibrosis.

Introduction

Hepatitis C virus (HCV) infection is a major health problem affecting >71.1 million people worldwide, leading to chronic hepatitis, liver cirrhosis, and hepatocellular carcinoma (HCC) (WHO, 2017). The advancement of HCV treatment in terms of the development of direct-acting antiviral agents (DAAs) has evoked international interest in the global elimination of HCV. In 2017, the World Health Organization set targets to eliminate viral hepatitis as a public threat worldwide by 2030 by achieving a 90% diagnosis rate, an 80% treatment rate, and a 65% reduction in the mortality rate. In patients with chronic HCV, the new DAAs can achieve a sustained virological response (SVR) rate >90%. International guidelines, for example, the 2018 European Association for the Study of the Liver and 2018 American Association for the Study of Liver Diseases guidelines recommend DAAs as the first-line treatment. The role of pegylated interferon (PEG-IFN) plus ribavirin has continued to diminish. However, up to 80% of the global HCV burden resides in low- and middle-income countries, including those in Southeast Asia, the Middle East, and North Africa. Due to the high cost and the lack of availability of DAAs, PEG-IFN-based therapy remains the treatment of choice in these countries (Jayasekera et al., 2014; Mohd Hanafiah et al., 2013; Zoulim et al., 2015). The current Asian Pacific Association for the Study of the Liver guidelines on the treatment of HCV infection continue to recommend PEG-IFN and ribavirin as first-line therapy in resource-limited countries where DAAs are unavailable (Omata et al., 2016).

Current evidence shows that in addition to playing roles in supporting calcium absorption and bone metabolism, vitamin D (VD) plays several important roles in immunomodulation, regulation of cellular proliferation, differentiation, and apoptosis (Holick, 2007; Penna et al., 2005; Von Essen et al., 2010). Several studies have reported associations between VD deficiency and risk of cancer, congestive heart failure, insulin resistance, and autoimmune diseases (Feskanich et al., 2004; Giovannucci et al., 2006; Munger et al., 2006). The liver is a crucial organ in VD synthesis as it is the site of the enzymatic conversion of the inactive form of VD to 25-dihydroxyVD. VD deficiency was found in 70% of patients with chronic liver disease regardless of the etiology, and 22% had severe VD deficiency (Arteh, Narra & Nair, 2010). Patients with chronic HCV infection had lower serum VD levels than sex- and age-matched healthy controls (Petta et al., 2010). In terms of clinical outcomes, low VD level has been reported to be independently related to advanced liver fibrosis and high necroinflammatory activity in chronic HCV patients (Dadabhai et al., 2017; Petta et al., 2010). Two large meta-analyses reported a negative association between VD level and SVR in chronic HCV patients treated with PEG-IFN therapy (Garcia-Alvarez et al., 2014; Villar et al., 2013).

The vitamin D receptor (VDR) is a nuclear hormone receptor that can act as a ligand-induced transcription factor. VDR binds to the active form of VD and thereby mediates its effect (Keane et al., 2018). The receptor is encoded by the VDR gene, which is located on chromosome 12q. The gene has a promoter, regulatory regions, and exons 2–9, which span over 100 kb (Deeb, Trump & Johnson, 2007; Uitterlinden et al., 2004). Using different restriction endonucleases for the BsmI, Tru9I, ApaI, and TaqI sites (to cleave the DNA at the 3′ end) and FokI (to cleave the DNA in exon 2), multiple VDR polymorphisms have been explored (Uitterlinden et al., 2004). The bAt (CCA) haplotype is a common genetic variant of the VDR gene, comprising the following three polymorphisms at the 3′ end of the gene: BsmI rs1544410 C, ApaI rs7975232 C, and TaqI rs731236 A, which are in strong linkage disequilibrium. Recent research shows that VDR genetic variations lead to susceptibility and chronicity regarding HCV infection (Wu et al., 2016). In addition, VDR polymorphisms may be related to the response to PEG-IFN and ribavirin therapy in chronic HCV patients. However, there have been conflicting results regarding these relationships in previous studies (Baur et al., 2012b; Garcia-Martin et al., 2013; Hung et al., 2016; Shaker et al., 2016). This study aims to investigate whether the common VDR polymorphisms are associated with the response to PEG-IFN-based therapy and advanced liver fibrosis in patients with chronic HCV infection.

Materials and Methods

Patients

This study included Thai patients with chronic HCV infection at Chulalongkorn University hospital (Bangkok, Thailand) and Srinagarind hospital (Khon Kaen, Thailand) from June 2012 to December 2013. All patients had positive anti-HCV antibody and detectable HCV RNA. They were treated with PEG-IFN and ribavirin based on standard recommendations (European Association for the Study of the Liver, 2011; Ghany et al., 2009). The exclusion criteria were co-infection with hepatitis B virus or human immunodeficiency virus, decompensated cirrhosis, or prior liver transplantation. Baseline characteristics were recorded, and biochemical and virological tests were conducted at baseline, during treatment and at 24 weeks after treatment. Alcohol consumption was defined as at least three standard drinks per week. The Fibrosis-4 (FIB-4) score (based on age, aspartate and alanine aminotransferase levels, and platelet count) was used to assess liver fibrosis. Advanced liver fibrosis was defined as FIB-4 score >3.25 (Vallet-Pichard et al., 2007).

The study followed the principles of the Declaration of Helsinki and was approved by the local Institutional Review Board (IRB) committee of the Faculty of Medicine, Chulalongkorn University (IRB number 562/54) and Khon Kaen University (HE561177). Written informed consent was obtained from each participant.

Virological testing

The quantitative serum HCV RNA level was evaluated using the real-time polymerase chain reaction (RT-PCR) COBAS® Taqman® HCV test (Roche Diagnostics, Basel, Switzerland). HCV genotyping was performed using the INNO-LiPA HCV II assay (Innogenetics, Ghent, Belgium).

Genotyping

Genotyping of the following six single-nucleotide polymorphisms (SNPs) was performed: the interleukin 28B (IL28B; also known as interferon lambda 3 [IFNl3]) C > T (rs12979860) SNP and five VDR SNPs, comprising FokI T > C (rs2228570), BsmI C > T (rs1544410), Tru9I G > A (rs757343), ApaI C > A (rs7975232), and TaqI A > G (rs731236). DNA was extracted from 100 μL of peripheral blood leukocytes using a standard phenol-chloroform protocol and then kept at –80 °C. Next, two μL DNA was subjected to PCR (total volume, 25 μL) using Perfect Taq Plus MasterMix (5 PRIME GmbH, Hamburg, Germany). The PCR-specific probes and conditions are summarized in Table S1. To assess the IL28B SNP, a sequencing method was used (First BASE Laboratories, Selangor, Malaysia). To assess the five VDR SNPs, restriction fragment length polymorphism assays were conducted. Subsequently, 2% agarose gel electrophoresis was used to assess the DNA fragments. The separated DNA was viewed under ultraviolet light after staining with ethidium bromide.

Three of the SNPs located at the 3′ end of the VDR gene (BsmI, ApaI, and TaqI) are in strong linkage disequilibrium, and the bAt (CCA) haplotype involves BsmI rs1544410 C, ApaI rs7975232 C, and TaqI rs731236 A.

Statistical analysis

Statistical analysis was performed using SPSS version 22.0 (IBM Corp., Armonk, NY, USA). Categorical data are expressed as number (percentage), and the differences between groups were compared using the chi-square test. Continuous data are expressed as mean ± standard deviation, and the differences between groups were compared using Student’s t-test and the Mann–Whitney U-test. The effects of pre-treatment factors and the SNPs on the response to PEG-IFN-based therapy (in terms of SVR) and the presence of advanced liver fibrosis were investigated using univariate and stepwise multivariate logistic regression analyses. A p-value <0.05 was considered statistically significant. The chi-square test was used to verify whether the genotype frequencies related to the SNPs in patients with and without SVR were in accordance with the Hardy–Weinberg assumption.

Results

Patient characteristics

A total of 554 Thai patients with chronic HCV infection were enrolled. There were 365 men (65.9%) and the mean age was 50.9 ± 9.2 years. A total of 334 patients (60.3%) achieved SVR, 255 patients (46%) were infected with HCV genotype 1, and 176 patients (34.8%) had advanced liver fibrosis. Table 1 shows the participants’ baseline demographic and laboratory data according to treatment response at 24 weeks after PEG-IFN discontinuation. Compared to patients with poor response (in terms of lack of an SVR), patients who achieved SVR were older, had lower pre-treatment serum HCV RNA levels, and were less likely to have HCV genotype 1, advanced liver fibrosis, and the unfavorable IL28B rs12979860 CT/TT genotypes.

Table 1 Baseline patient characteristics according to response to PEG-IFN-based therapy.

	Non-SVR (n = 220)	SVR (n = 334)	p-value	
Female, n (%)	71 (32.3%)	118 (35.3%)	0.46	
Age (years), mean ± SD	52.1 ± 8.0	50.1 ± 9.8	0.01	
Body mass index (kg/m2), mean ± SD	24.6 ± 3.4	24.6 ± 3.6	0.96	
Alcohol drinking, n (%)	134 (69.8%)	142 (61.7%)	0.1	
Diabetes mellitus, n (%)	51 (26.2%)	53 (22.7%)	0.41	
Genotype, n (%)	
 1	122 (55.5%)	133 (39.8%)	<0.001	
 2	0	1 (0.3%)		
 3	85 (38.6%)	160 (47.9%)		
 6	13 (5.9%)	40 (12%)		
HCV RNA (logIU/mL), mean ± SD	6.05 ± 0.61	5.8 ± 0.8	0.002	
ALT (U/L), mean ± SD	107.5 ± 166.5	100.1 ± 74.0	0.49	
Advanced liver fibrosis, n (%)	82 (41.2%)	94 (30.6%)	0.02	
IL28B rs12979860, n (%)	
 CC	148 (67.3%)	290 (86.8%)	<0.001	
 CT	69 (31.4%)	38 (11.4%)		
 TT	3 (1.4%)	6 (1.8%)		

Prevalence of VDR polymorphisms and bAt (CCA) haplotype and their associations with response to PEG-IFN-based therapy

The frequencies of the VDR genotypes and the bAt (CCA) haplotype and their associations with response to PEG-IFN-based therapy are shown in Table 2. The genotypic frequencies of the SNPs were in Hardy–Weinberg equilibrium (p > 0.05) except for Tru9I (rs757343). The genotypic frequencies of the SNPs were not different between patients with and without SVR.

Table 2 Frequencies of the VDR genotypes and the bAt haplotype in Thai patients with chronic hepatitis C infection treated with PEG-IFN.

	All patients (n = 554)	Non-SVR (n = 220)	SVR (n = 334)	Odds ratio (95% CI)	p-value	
FokI rs2228570	
 TT	116 (20.9%)	51 (23.2%)	65 (19.5%)	0.80 [0.53–1.21]	0.29	
 TC	271 (48.9%)	105 (47.7%)	166 (49.7%)			
 CC	167 (30.2%)	64 (29.1%)	103 (30.8%)			
BsmI rs1544410	
 CC	453 (81.8%)	181 (82.3%)	272 (81.4%)	0.95 [0.61–1.47]	0.80	
 CT	94 (17.0%)	36 (16.4%)	58 (17.4%)			
 TT	7 (1.3%)	3 (1.4%)	4 (1.2%)			
Tru9I rs757343	
 GG	326 (58.8%)	136 (61.8%)	190 (56.9%)	0.82 [0.58–1.15]	0.25	
 GA	197 (35.6%)	74 (33.6%)	123 (36.8%)			
 AA	31 (5.6%)	10 (4.5%)	21 (6.3%)			
ApaI rs7975232	
 CC	252 (45.5%)	106 (48.2%)	146 (43.7%)	0.84 [0.59–1.18]	0.30	
 CA	240 (43.3%)	95 (43.2%)	145 (43.4%)			
 AA	62 (11.2%)	19 (8.6%)	43 (12.9%)			
TaqI rs731236	
 AA	477 (86.1%)	197 (89.5%)	280 (83.8%)	0.61 [0.36–1.02]	0.06	
 AG	68 (12.3%)	23 (10.5%)	45 (13.5%)			
 GG	9 (1.6%)	0	9 (2.7%)			
bAt (CCA) haplotype	486 (87.7%)	201 (91.4%)	285 (85.3%)	1.82 [1.04–3.18]	0.03	

The FokI, BsmI, Tru9I, ApaI, and TaqI polymorphisms were not associated with response to PEG-IFN-based therapy. However, the bAt (CCA) haplotype was significantly associated with poor response to PEG-IFN-based therapy. Overall, 41.4% of patients with the bAt (CCA) haplotype were poor responders, resulting in an OR of 1.82 (95% CI [1.04–3.18], p = 0.03) when compared to patients with other haplotypes (27.9%).

Factors associated with response to PEG-IFN-based therapy

Table 3 shows the univariate and multivariate analysis results of the effects of baseline variables on the response to PEG-IFN-based therapy. Based on univariate analysis, advanced age, HCV genotype 1, high pre-treatment HCV RNA level, advanced liver fibrosis, IL28B rs12979860 CT/TT, and the bAt (CCA) haplotype were significantly associated with poor response to PEG-IFN-based therapy. Stepwise multivariate regression analysis showed that the IL28B rs12979860 CT/TT genotypes (OR = 3.44, 95% CI [2.12–5.58], p < 0.001), the bAt (CCA) haplotype (OR = 2.02, 95% CI [1.04–3.91], p = 0.03), pre-treatment HCV RNA level (logIU/mL; OR = 1.73, 95% CI [1.31–2.28], p < 0.001), advanced liver fibrosis (OR = 1.68, 95% CI [1.10–2.58], p = 0.02), and HCV genotype 1 (OR = 1.59, 95% CI [1.07–2.37], p = 0.02) were independent baseline predictors of poor response to PEG-IFN-based therapy.

Table 3 Univariate and multivariate regression analyses of factors associated with poor response to pegylated interferon-based therapy in patients with chronic HCV infection.

	Univariate analysis	Multivariate analysis	
	OR (95% CI)	p-value	OR (95% CI)	p-value	
Female	0.87 [0.61–1.25]	0.46			
Age	1.02 [1.00–1.04]	0.01	1.02 [0.99–1.00]	0.11	
Body mass index	1.00 [0.95–1.06]	0.97			
Alcohol drinking	1.43 [0.95–2.15]	0.1			
Diabetes mellitus	1.20 [0.77–1.87]	0.41			
Genotype 1	1.89 [1.33–2.66]	<0.001	1.59 [1.07–2.37]	0.02	
HCV RNA (logIU/mL)	1.47 [1.15–1.88]	0.002	1.73 [1.31–2.28]	<0.001	
ALT (U/L)	1.00 [0.99–1.00]	0.50			
Advanced liver fibrosis	1.59 [1.10–2.30]	0.02	1.68 [1.10–2.58]	0.02	
IL28B rs12979860 CT/TT	3.21 [2.10–4.90]	<0.001	3.44 [2.12–5.58]	<0.001	
bAt haplotype	1.82 [1.04–3.18]	0.03	2.02 [1.04–3.91]	0.03	

Comparison between bAt (CCA) and other haplotypes

The vitamin D receptor is expressed in various cell types in the liver, including hepatic stellate cells, Kupffer cells, endothelial cells, and hepatocytes, and upregulated during hepatic injury. It involves in immune regulations. Accordingly, it might affect clinical outcomes of chronic HCV patients treated with PEG-IFN/ribavirin. Several important factors were compared between the chronic HCV patients with bAt (CCA) and other haplotypes to identify the associations with the bAt (CCA) haplotype (Table 4). Among the participants, 486 (87.7%) had the bAt (CCA) haplotype. There were no differences in pre-treatment HCV RNA level, advanced liver fibrosis, or rapid or early virological response between patients with bAt (CCA) and patients with other haplotypes. However, patients with the bAt (CCA) haplotype were more likely to have unfavorable IL28B rs12979860 CT/TT genotypes, and they had lower pre-treatment alanine aminotransferase levels than patients with other haplotypes.

Table 4 Baseline characteristics, virological factors, and liver fibrosis stage in accordance to the bAt (CCA) haplotype.

	CCA haplotype (n = 486)	Other haplotypes (n = 68)	p-value	
Pre-treatment HCV RNA level (log IU/mL), mean ± SD	5.93 ± 0.77	5.87 ± 0.69	0.57	
Pre-treatment ALT, mean ± SD	97.3 ± 68.0	143.5 ± 285.9	0.004	
Advanced liver fibrosis, n (%)	158 (35.5%)	18 (29.5%)	0.36	
Rapid virological response, n (%)	257 (65.6%)	37 (71.2%)	0.42	
Early virological response, n (%)	371 (88.8%)	48 (87.3%)	0.75	
IL28B rs12979860 CT/TT genotypes, n (%)	93 (19.1%)	23 (33.8%)	0.005	

Associations between VDR polymorphisms and both advanced liver fibrosis and HCV RNA level

A total of 506 patients (91.3%) had pre-treatment laboratory data for calculating FIB-4 score. Of these, 176 patients (34.8%) had FIB-4 score >3.25 and were thus diagnosed with advanced liver fibrosis. Figure 1 shows the prevalence of advanced liver fibrosis among patients with each VDR genotype. Chronic HCV patients with the FokI rs2228570 TT/TC genotypes (38.6%) were more likely to have advanced liver fibrosis compared to patients with the CC genotype (25.8%, p = 0.006). Pre-treatment HCV RNA level was not significantly different among patients who had different VDR genotypes, as shown in Fig. 2.

Figure 1 Association between advanced liver fibrosis and VDR polymorphisms in patients with chronic HCV infection.

(A) FokI rs2228570 T > C, (B) BsmI rs1544410 C > T, (C) Tru9I rs757343 G > A, (D) ApaI rs7975232 C > A, (E) TaqI rs731236 A > G.

Figure 2 Baseline serum HCV RNA according to VDR polymorphisms in patients with chronic HCV infection.

(A) FokI rs2228570 T > C, (B) BsmI rs1544410 C > T, (C) Tru9I rs757343 G > A, (D) ApaI rs7975232 C > A, (E) TaqI rs731236 A > G.

Factors associated with advanced liver fibrosis

Univariate and multivariate analysis results for advanced liver fibrosis are shown in Table 5. Based on the univariate analysis, advanced liver fibrosis was associated with age ≥55 years (p < 0.001) and FokI TT/TC genotypes (p = 0.006). Factors with p < 0.1 in the univariate analysis were included in the multivariate model. Based on the multivariate analysis, age ≥55 years (OR = 2.25; 95% CI [1.54–3.32], p < 0.001) and FokI TT/TC genotypes (OR = 1.63; 95% CI [1.06–2.51], p = 0.03) were independent predictors of advanced liver fibrosis. The BsmI, Tru9I, ApaI, and TaqI genotypes, and the bAt (CCA) haplotype were not associated with advanced liver fibrosis.

Table 5 Univariate and multivariate regression analyses of factors associated with advanced liver fibrosis in patients with chronic HCV infection.

	Univariate analysis	Multivariate analysis	
	OR (95% CI)	p-value	OR (95% CI)	p-value	
Age ≥ 55 years	2.38 [1.62–3.49]	<0.001	2.25 [1.54–3.32]	<0.001	
Female	1.37 [0.93–2.00]	0.11			
Body mass index	1.01 [0.96–1.07]	0.72			
Alcohol consumption	1.25 [0.79–1.98]	0.35			
HCV genotype 1	0.75 [0.52–1.09]	0.75			
IL28B rs12979860 CT/TT genotypes	0.94 [0.60–1.47]	0.78			
FokI rs2228570 TT/TC genotypes	1.81 [1.18–2.75]	0.006	1.63 [1.06–2.51]	0.03	
BsmI rs1544410 GG genotype	1.04 [0.64–1.69]	0.88			
Tru9l rs757343 GG genotype	1.00 [0.69–1.45]	1.00			
Apal rs7975232 GG genotype	0.90 [0.62–1.30]	0.57			
TaqI rs731236 TT genotype	0.90 [0.52–1.54]	0.70			
bAt (CCA) haplotype	1.32 [0.73–2.36]	0.36			

Discussion

The main findings are that the FokI rs2228570 TT/TC genotypes are independently associated with an increased risk of advanced liver fibrosis in Thai chronic HCV patients. Additionally, the VDR bAt (CCA) haplotype was independently associated with poor response to PEG-IFN and ribavirin in patients with chronic HCV infection. Interestingly, these associations did not depend on the unfavorable IL28B rs12979860 CT/TT genotypes, HCV genotype, or pre-treatment HCV viral load. This study provides evidence indicating the important effects of VDR polymorphisms on clinical outcomes in patients with chronic HCV infection.

Vitamin D receptor acts as a ligand-induced transcription factor that binds to 1,25-dihydroxyVD and exerts its effects by regulating the expression of >900 genes in target tissues (Kato, 2000). Recent studies have indicated that 1,25-dihydroxyVD and VDR are important regulators of both the innate and adaptive immune response (Khammissa et al., 2018; Rosen et al., 2012). VDR is expressed in almost all immune cells including B cells, activated T lymphocytes, neutrophils, natural killer cells, and antigen-presenting cells (Bhalla et al., 1983; Provvedini et al., 1983). The 1,25-dihydroxyVD/VDR signaling pathway can activate monocytes, inhibit lymphocyte proliferation, and prevent the differentiation of dendritic cell precursors into antigen-presenting cells (Berer et al., 2000). In addition, 1,25-dihydroxyVD is able to suppress IFN-γ transcription via the binding of VDR to a silencer region in the IFN-γ gene promoter (Saggese et al., 1989). Genetic variations of the VDR gene can result in a dysfunctional receptor that subsequently affects the function of VD. VDR polymorphisms have been implicated in susceptibility to a variety of autoimmune diseases and cancers in a genome-wide association study and meta-analysis (Raimondi et al., 2009; Ramagopalan et al., 2010). Interestingly, VDR variants regulate the biological effects of VD independently of the serum 1,25-dihydroxyVD level (Khammissa et al., 2018).

Regarding the association between VDR and response to PEG-IFN-based therapy in chronic HCV infection, a recent in vitro study reported that 1,25-dihydroxyVD promotes the inhibitory effect of IFN-α on HCV replication by enhancing the expression of IFN-stimulated genes (Beilfuss et al., 2015). The crosstalk between VDR and IFN-α signaling may help to better understand the underlying mechanisms in clinical studies of HCV infection. The results from the current study showed that the VDR bAt (CCA) haplotype was associated with poor response to PEG-IFN-based therapy in Thai patients with chronic HCV infection. Although this association has been reported in several previous studies, the findings have been conflicting. Baur et al. (2012b) and Garcia-Martin et al. (2013) reported that Caucasian patients with chronic HCV infection with the bAt (CCA) haplotype had an impaired response to PEG-IFN and ribavirin. In contrast, Hung et al. (2016) did not find an association between the bAt (CCA) haplotype and antiviral response to PEG-IFN therapy in 139 Taiwanese patients with chronic HCV genotype-1 infection. The possible reason for the discordant results between the two studies in Asian chronic HCV patients (i.e., our study and the Hung et al. (2016) study) may be the lower prevalence of the bAt haplotype in the previous study (54.7%) compared to that in our study (87.7%). The mechanism underlying the association between the bAt (CCA) haplotype and poor response to PEG-IFN is still unclear. It may be due to the effect of the haplotype on the immune response-related IFN signaling cascade (Ramagopalan et al., 2010), as we found no relationship between the bAt (CCA) haplotype and pre-treatment HCV RNA level or liver fibrosis stage. Additionally, our study did not find any relationships between the VDR SNPs and the response to PEG-IFN-based therapy. In contrast, previous studies reported negative associations between the response to PEG-IFN-based therapy and both the FokI T allele (Garcia-Martin et al., 2013; Barchetta et al., 2012; Baur et al., 2012a) and the TaqI G allele (Baur et al., 2012a) in patients with chronic HCV infection.

With regard to the relationship between VDR polymorphisms and clinical outcomes, an in vitro analysis showed that VDR ligands inhibited transforming growth factor (TGF)-β1-induced hepatic stellate cell activation and reduced liver fibrosis, while, in a mouse model, genetic knockout of VDR expression led to spontaneous liver fibrosis (Ding et al., 2013). The response of human hepatic stellate cells to TGF-β1 and VD partially depends on the VDR polymorphisms (Beilfuss et al., 2015). In patients with chronic HCV genotype 1 infection, low serum VD level is related to severe liver fibrosis (Petta et al., 2010). VDR is expressed in hepatic parenchymal and inflammatory cells of patients with chronic HCV infection, and low VDR expression is associated with high portal inflammation (Barchetta et al., 2012). The current study showed that the VDR FokI rs2228570 TT/TC genotypes and age ≥55 years were independent risk factors for advanced liver fibrosis in Thai patients with chronic HCV infection. The BsmI, Tru9I, ApaI, and TaqI polymorphisms, and the bAt (CCA) haplotype, were not associated with advanced liver fibrosis or pre-treatment HCV RNA level in our cohort. Previous research reported that VDR variants were related to decreased HCV infection susceptibility in a Chinese population (Wu et al., 2016). A cohort study of Swiss chronic HCV patients showed the bAt (CCA) haplotype was associated with rapid fibrosis progression and cirrhosis (Baur et al., 2012a). Additionally, BsmI and TaqI polymorphisms were associated with liver fibrosis in a Brazilian cohort (Scalioni et al., 2018). Moreover, in Taiwanese patients with chronic HCV infection, the bAt (CCA) haplotype, ApaI CC genotype, and TaqI AA genotype were associated with increased HCV RNA levels compared to other genotypes/haplotypes (Hung et al., 2016). Furthermore, the ApaI CC genotype was an independent factor for the development of HCC in a Taiwanese cohort with chronic HCV infection (Hung et al., 2014).

The FokI polymorphism restriction site is located in exon 2 in the 5′ coding region of VDR. This polymorphism leads to a T > C (threonine to cysteine) substitution and the generation of a protein shortened by three amino acids, which makes the protein less functionally active than the wild type (Van Etten et al., 2007). This polymorphism has been implicated in the response to PEG-IFN therapy and several chronic liver diseases including autoimmune hepatitis and HCC in patients with chronic HBV infection (Mostafa-Hedeab et al., 2018; Vogel, Strassburg & Manns, 2002; Yao et al., 2013). The present study found an association between the FokI polymorphism and advanced liver fibrosis in patients with chronic HCV infection. The FokI polymorphism genotypic frequencies in a healthy Thai population have been reported to be 15.7% for TT, 43.6% for TC, and 40.7% for CC (Sangkaew, Nuinoon & Jeenduang, 2018). These frequencies are consistent with the frequencies in our study of 20.9% for TT, 48.9% for TC, and 30.2% for CC in Thai patients with chronic HCV infection.

Our study had several limitations. First, we did not investigate the relationship between the baseline serum VD level and response to PEG-IFN-based therapy because it is influenced by many confounding factors such as sunlight exposure, nutritional status, and liver function. In addition, VDR variants can modulate their effects independently of serum VD status (Uitterlinden et al., 2004). Second, our study was a retrospective study, and pre-treatment serum samples were not collected for most of the participants. However, we still attempted to assess associations between the VDR variants and clinical outcomes in patients with chronic HCV infection. Third, the combination of HCV infection with either alcoholic liver disease or non-alcoholic liver disease (NAFLD) could accelerate the progression of liver fibrosis, but we did not exclude patients with alcoholic liver disease or NAFLD. However, to identify independent associations between the studied polymorphisms and advanced fibrosis in chronic HCV patients, we used a stepwise multivariate analysis to adjust for confounding factors such as alcohol consumption, body mass index, and type 2 diabetes.

Conclusions

The present study demonstrates an association between the VDR bAt (CCA) haplotype and poor response to PEG-IFN plus ribavirin therapy and associations between the VDR FokI rs2228570 TT/TC genotypes and advanced liver fibrosis in Thai patients with chronic HCV infection. These results provide helpful clinical information for understanding the causative effects of VDR polymorphisms on clinical outcomes. Further studies are required to elucidate the detailed molecular mechanisms.

Supplemental Information

Supplemental Information 1 Genes, single nucleotide polymorphism numbers, primer sequences and polymerase chain reaction conditions.

Click here for additional data file.

Supplemental Information 2 VDR1.

Click here for additional data file.

Supplemental Information 3 Codebook for VDR1 database.

Click here for additional data file.

We would like to thank the staff of the Division of Gastroenterology and Hepatology (Chulalongkorn University), Center of Excellence in Liver Diseases, King Chulalongkorn Memorial Hospital, Thai Red Cross Society), Liver Fibrosis and Cirrhosis Research Unit (Chulalongkorn University), and the Center of Excellence in Clinical Virology (Chulalongkorn University) for their technical assistance and clinical support.

Additional Information and Declarations

Competing Interests

Author Contributions

Human Ethics

Field Study Permissions

Data Availability

The authors declare that they have no competing interests.

Kessarin Thanapirom conceived and designed the experiments, analyzed the data, contributed reagents/materials/analysis tools, prepared figures and/or tables, authored or reviewed drafts of the paper, approved the final draft.

Sirinporn Suksawatamnuay performed the experiments, analyzed the data, approved the final draft.

Wattana Sukeepaisarnjaroen contributed reagents/materials/analysis tools, approved the final draft.

Pisit Tangkijvanich contributed reagents/materials/analysis tools, approved the final draft.

Panarat Thaimai performed the experiments, approved the final draft.

Rujipat Wasitthankasem performed the experiments, approved the final draft.

Yong Poovorawan contributed reagents/materials/analysis tools, approved the final draft.

Piyawat Komolmit conceived and designed the experiments, contributed reagents/materials/analysis tools, prepared figures and/or tables, authored or reviewed drafts of the paper, approved the final draft.

The following information was supplied relating to ethical approvals (i.e., approving body and any reference numbers):

The study protocol was approved by the Institutional Review Board of the Faculty of Medicine, Chulalongkorn University (IRB number 562/54) and Khon Kaen University (HE561177).

The following information was supplied relating to field study approvals (i.e., approving body and any reference numbers):

The study protocol was approved by the Institutional Review Board of the Faculty of Medicine Khon Kaen University (HE561177).

The following information was supplied regarding data availability:

Raw data are available in the Supplemental File.

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
