# Peer review of "Genetic associations of vitamin D receptor polymorphisms with advanced liver fibrosis and response to pegylated interferon-based therapy in chronic hepatitis C"

_PeerJ, doi:10.7717/peerj.7666_

## Round 0.1 · original submission · Major Revisions

Please provide point-to-point responses to the reviewer comments, and a tracked-changes copy of the revision.Thank you!

·

Basic reporting

.

Experimental design

.

Validity of the findings

.

Additional comments

Title:
the title (Genetic Associations of Vitamin D Receptor Polymorphisms 2 with Advanced Liver Fibrosis and Response to Pegylated 3 Interferon-based Therapy in Chronic Hepatitis C) is informative with clear and relevant aim.

Introduction:
• The research question justified given what is already known about the topic.
• Please add the exclusion criteria and summarize the inclusion ones in a points rather than a paragraph.

Methods:
● The process of subject selection clear.
● Variables defined and measured appropriately.
● We need enough details in order to replicate the study!


Results:
● The data presented in an appropriate way.
● Table 1: Table 1: Baseline patient characteristics according to response to PEG-IFN2 based therapy (no significant selection bias).
● There is a huge selection bias in Table 3: Baseline characteristics, virological factors and liver fibrosis stage in 2 accordance to the bAt (CCA) haplotype! Please try to make a comment on that to support your study because it might weaken your study significantly.
● Five hundred and ten patients had data on liver fibrosis stage. Of these, 269 underwent liver biopsy before treatment with PEG-IFN and ribavirin=Please make a referral for any adverse events during the liver biopsies?



Discussion& Conclusion:
● The conclusions answer the aims of the study.
● The conclusions supported by references or results.

·

Basic reporting

1. For the treatment of HCV, the 2018 EASL and the 2018 AASLD recommendations for treating hepatitis C both consider DAAs as the first-line treatment. Therefore, did the authors consider the clinical significance and necessity of pegylated interferon-based therapy in the context of these recent recommendations?

Experimental design

2. Alcohol and obesity are also very important factors affecting the degree of liver inflammation, especially alcoholic liver disease and non-alcoholic fatty liver disease. Researchers did not exclude the patients with HCV who also had alcoholic liver disease or NAFLD. However, the use of alcohol as a variable could affect the conclusions.

Validity of the findings

3. Line 209, The advanced liver fibrosis was assessed using two criteria: liver histology and FIB4 index. Liver histology is the gold standard for evaluating the degree of liver fibrosis, while FIB4 is a non-invasive marker. Liver histology and FIB4 are essentially different. A potential measurement bias may be introduced if some assessed using histology and others using FIB4. Therefore, all patients should be assessed for liver fibrosis using either of histology or FIB4.

Additional comments

1. In results, lines 180-187, why not make a table which would show results clearer and more concise?
2. Line 210, the P value of being male showed 0.08, which is not statistically significant. Please check it.
3. In the introduction and discussion, there were a lot of repetition. Would authors mind to reduce them appropriately?
4. There were a lot of ambiguity in the article. It is suggested that native English experts revise the manuscript.

---

## Round 0.2 · accepted · Accept

Thanks for thorough responses to the reviewer comments!

·

Basic reporting

1. For the treatment of HCV, the 2018 EASL and the 2018 AASLD recommendations for treating hepatitis C both consider DAAs as the first-line treatment. Therefore, did the authors consider the clinical significance and necessity of pegylated interferon-based therapy in the context of these recent recommendations?
It has been addressed.

Experimental design

1.Alcohol and obesity are also very important factors affecting the degree of liver inflammation, especially alcoholic liver disease and non-alcoholic fatty liver disease. Researchers did not exclude the patients with HCV who also had alcoholic liver disease or NAFLD. However, the use of alcohol as a variable could affect the conclusions.
It has been addressed.

Validity of the findings

3. Line 209, The advanced liver fibrosis was assessed using two criteria: liver histology and FIB4 index. Liver histology is the gold standard for evaluating the degree of liver fibrosis, while FIB4 is a non-invasive marker. Liver histology and FIB4 are essentially different. A potential measurement bias may be introduced if some assessed using histology and others using FIB4. Therefore, all patients should be assessed for liver fibrosis using either of histology or FIB4.
It has been addressed.

Additional comments

1. In results, lines 180-187, why not make a table which would show results clearer and more concise?
It has been addressed.
2. Line 210, the P value of being male showed 0.08, which is not statistically significant. Please check it.
It has been addressed.
3. In the introduction and discussion, there were a lot of repetition. Would authors mind to reduce them appropriately?
It has been addressed.
4. There were a lot of ambiguity in the article. It is suggested that native English experts revise the manuscript.
It has been addressed.